# AMS-02 antiprotons and dark matter:
# Trimmed hints and robust bounds

**Francesca Calore**[1⋆], **Marco Cirelli**[2†], **Laurent Derome**[3‡], **Yoann Génolini**[1,4∘],
**David Maurin**[3§], **Pierre Salati**[1¶] **and Pasquale D. Serpico**[1∥]

**1** LAPTh, USMB, CNRS, F-74940 Annecy, France
**2** LPTHE, CNRS & Sorbonne University, 4 place Jussieu, Paris, France
**3** LPSC, Univ. Grenoble Alpes, CNRS, 53 avenue des Martyrs, F-38000 Grenoble, France
**4** Niels Bohr International Academy & Discovery Center, Niels Bohr Institute,
University of Copenhagen, Blegdamsvej 17, DK-2100 Copenhagen, Denmark

⋆ lapth.cnrs.fr , † cirelli@lpthe.jussieu.fr , ‡ Derome@lpsc.in2p3.fr ,
∘ genolini@lapth.cnrs.fr , § dmaurin@lpsc.in2p3.fr ,
¶ salati@lapth.cnrs.fr , ∥ serpico@lapth.cnrs.fr

## Abstract

Based on 4 yr AMS-02 antiproton ($\bar{p}$) data, we present bounds on the dark matter (DM) annihilation cross section vs. mass for some representative final state channels. We use recent cosmic-ray propagation models, a realistic treatment of experimental and theoretical errors, and an updated calculation of input $\bar{p}$ spectra based on a recent release of the PYTHIA code. We find that reported hints of a DM signal are statistically insignificant; an adequate treatment of errors is crucial for credible conclusions. Antiproton bounds on DM annihilation are among the most stringent ones, probing thermal DM up to the TeV scale. The dependence of the bounds upon propagation models and the DM halo profile is also quantified. A preliminary estimate reaches similar conclusions when applied to the 7 years AMS-02 dataset, but also suggests extra caution as for possible future claims of DM excesses.

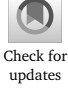

# 1 Introduction

Cosmic rays (CRs), essentially Galactic non-thermal hydrogen and helium nuclei with traces of electrons and heavier nuclei, have historically provided the earliest indication of a striking matter-antimatter asymmetry (for an overview of these earlier evidences, see [1]). Nonetheless, traces of antiprotons $\bar{p}$ (and even more rare antinuclei [2]), are expected to be produced via CR collisions with the interstellar medium (ISM). Since the discovery of the $\bar{p}$ component over 40 years ago [3], antiprotons have been recognized as an important tool for probing the CR propagation in the ISM (for reviews, see [4–7]). The theoretically appealing weakly interacting massive particles (WIMPs) are motivated dark-matter (DM) candidates, potentially yielding an additional *primary* source of antiprotons [8]. In the last couple of decades, antiprotons have provided one of the best tools to probe the WIMP DM parameter space (for a recent review, see [9]; see also [10–23]).

A few years ago, the AMS-02 mission published a $\bar{p}-$flux measurement with unprecedented precision and over a wide dynamical range [24]. These data have not only been used to set very stringent bounds on the DM annihilation cross section [11–13, 19, 20, 22, 23, 25], but have also been interpreted as hinting to a DM signal [14, 15, 21]. Recently, the AMS-02 collaboration has published updated flux measurements [26], whose impact on DM has only started to be explored in [22, 23, 27]. Now more than ever, fully exploiting the data requires to carefully handle both the observational and theoretical errors.

Over the past years, our group has embarked in a systematic analysis of these aspects involving primary spectra, spallation cross-sections, solar modulation, the halo size, and diffusion parameters [28–35]. These studies paved the way to a first analysis of $\bar{p}$ data, showing their consistency with a purely secondary origin (i.e. from the astrophysical collision process mentioned above) [36]. Moreover, in [33], we have derived constraints on the halo size, a crucial parameter affecting the primary DM signal. In the present work, we present a study of the $\bar{p}$ constraints on DM models, following our early assessment in [11], which was based on preliminary AMS-02 data and on pre-AMS-02 propagation models. Note that, since all the ingredients for the antiproton spectrum calculation and its uncertainties have been calibrated on the earlier AMS-02 data releases, for consistency we use the data of ref. [24] for the antiproton DM analysis. This also eases the comparison with most other results in the literature, obtained under the same hypotheses. Nonetheless, for completeness, in sec. 5.4 we present a first discussion of the robustness of our results in the light of the new flux measurements reported in [26].

This article is structured as follows. In Sec. 2, we present the DM input, notably the primary spectra (Sec. 2.1) and the halo profile adopted with its uncertainty (Sec. 2.2). Section 3 summarises our treatment and assumptions on the transport parameters, while in Sec. 4 we outline our methodology and the statistical approach used. In Sec. 5, we present our results. The best-fit models in presence of DM (Sec. 5.1) turn out to be statistically not-significant. In Sec. 5.2 we present the constraints in the $m_\chi - \langle \sigma v \rangle$ plane for a number of relevant channels, compare them with the literature, and discuss some sources of uncertainties (halo, propagation model). A comparison with some multimessenger bounds is presented in Sec. 5.3. In Sec. 5.4, we present a first assessment of the impact of the larger dataset published in [26] on the DM analysis. Conclusions with some perspectives are reported in Sec. 6. Details on the propagation model and solution adopted (already described elsewhere) are reported for completeness in the appendix A.

## 2 Dark matter inputs

### 2.1 Spectra

In order to span the range of representative antiproton spectra from DM annihilation, we consider the following channels:

$$b\bar{b}, \, W^+W^-, \, \mu^+\mu^-, \, q\bar{q}, \, hh, \tag{1}$$

which are kinematically open whenever $m_{\mathrm{DM}} > m_i$, with $i = (b, W, \mu, q, h)$ denoting the annihilation products and $m_i$ their mass. The $b\bar{b}$ channel is representative of hadronic annihilations. It is usually considered among the benchmark cases since in many models the branching ratio into this channel is sizable (e.g. if DM couples according to the mass) and it is 'always' open (since typically $m_{\mathrm{DM}} > m_b$ for WIMP-like DM). Similarly, the $W^+W^-$ channel is representative of gauge boson annihilations. The $q\bar{q}$ channel stands for the light quarks ($u, d, s$ and, to a certain extent $c$): they have a very similar spectrum, somewhat distinct from the one of heavier quarks, hence it is worth investigating it separately from the $b\bar{b}$ case. The $hh$ channel, where $h$ is the Higgs boson, has a spectrum which is intermediate between the hadronic and gauge boson one. Finally, the $\mu^+\mu^-$ channel is representative of leptonic annihilation channels, where antiprotons are produced by the phenomenon of electroweak radiation [37–43] in which a $W$ or $Z$ boson is radiated in the annihilation process and subsequently decays into hadronic states. This higher order process is relevant as soon as the DM mass is sufficiently large (above $\sim 100$ GeV), albeit it is only able to probe tree-level annihilation cross-sections that are $\sim 3$ orders of magnitude larger than in hadronic modes. Other channels like $gg$, $ZZ$, $t\bar{t}$, $\tau^+\tau^-$ and $e^+e^-$ have antiproton spectra very similar to $q\bar{q}$, $W^+W^-$, $b\bar{b}$, $\mu^+\mu^-$, respectively, and are therefore not worth investigating separately.

We take the spectra from an updated version of the Pppc4dmid set of tools [44]. Compared to the original 2010 version, this version is based on a more recent release of the Pythia Monte Carlo code [45], which includes in particular an almost complete treatment of electroweak emissions from outgoing leptons at all orders (they were included in the original 2010 version of Pppc4dmid only at first order).[1] For antiprotons, which are of interest here, the impact of the spectral updates is not dramatic, however. It can be quantified between a few percent and a factor two at most, across DM masses and channels.

---

[1]The only caveat is that, for the specific case of the $W^+W^-$ channel at large DM masses we still use the 2010 spectra, since the $W^* \to WZ$ and $Z^* \to WW$ splitting are not included yet in the Pythia version used.

Table 1: Parameters of the DM profiles.

| Profile | $\gamma$ | $r_s$ [kpc] | $\rho_s$ [$M_\odot/\mathrm{pc}^3$] |
|---|---|---|---|
| benchmark NFW | 1.0 | 19.6 | 0.00854 |
| cored | 0.0 | 7.7 | 0.08931 |
| contracted NFW | 1.25 | 27.2 | 0.00361 |

## 2.2 Dark matter profile

Concerning the dark matter distribution in the Galaxy, we adopt a standard 'generalized Navarro-Frenk-White (NFW)' profile

$$\rho_{\mathrm{DM}}(r) = \frac{\rho_s}{(r/r_s)^\gamma \, (1 + r/r_s)^{3-\gamma}} \, . \tag{2}$$

Here, as customary, $\rho_{\mathrm{DM}}$ is the DM density as a function of $r$, the galactocentric spherical coordinate. The parameter $\gamma$ sets the power-law scaling of the profile towards the inner Galaxy and at its outskirts. The halo structural parameters $\rho_s$ and $r_s$ set the overall density normalization and the scale length of the change of slope. They are determined in a number of different ways in the literature, notably by creating global models of the Milky Way and fitting them to the observational constraints derived from stellar kinematical data (see e.g. [46–50]).

In our main analysis we use the 'benchmark' NFW profile [51], which corresponds to $\gamma = 1$ and thus peaks as $1/r$ at the center and scales as $1/r^3$ at large $r$. However, we also explore the impact of two different choices: A cored profile ($\gamma = 0$), which features a constant central density at small $r$, and a 'contracted NFW' profile (choosing $\gamma = 1.25$), which features an enhanced peak towards the Galactic Centre, and is able to explain the spatial distribution of the so-called *Fermi*-LAT GeV excess, see ref. [52] for a review. We adopt the sets of parameters summarized in Table 1. These values are inspired by Ref. [48], which has been used in many past studies, including from our group, and therefore allow for an easier comparison than if we had chosen to adopt more recent determinations. With respect to Ref. [48], however, we modify slightly the normalization in order to ensure that all profiles predict the same DM local density $\rho_\odot = 0.385$ GeV/cm$^3$. This allows us to isolate the impact on our results of the *shape* of the profile, rather than its predicted local density (the latter falls in the range 0.3-0.6 GeV/cm$^3$, see [53] for a recent review). In turn, if the functional form of the profile is held fixed, it is easy to estimate the effect of varying the value of $\rho_\odot$: the $\bar{p}$ fluxes depend on $\rho_\odot^2$, and hence the limits are modified by the same factor. For all these sets, the distance of the Sun from the Galactic centre is fixed at $R_\odot = 8.20$ kpc, a distance in agreement with the recent determination of the position of the Galactic centre black hole with GRAVITY [54] or obtained from the use of 138 Gaia EDR3 globular clusters [55].

## 3 Propagation setup, parameters, and uncertainties

In this section we provide a brief description of the model and transport parameters entering our analysis, keeping in mind that more details can be found in App. A or in the original publications cited.

In short, the model is the 2-zone model (thin disc, thick halo of size $L$) with spatial diffusion $K$, convection $V_c$ (constant and perpendicular to the disc), and reacceleration $V_a$ (in the thin disc only). This model is analog to that introduced in [56] and already used in [10] to set constraints on WIMP DM from $\bar{p}$. Both the primary and secondary $\bar{p}$ flux calculations depend

on all the above model parameters. They are obtained from the analysis of Li/C, Be/B, and B/C data: whereas secondary-to-primary ratios allow us to extract $K(R)/L$, $V_c$, and $V_a$, the use of the radioactive clock $^{10}$Be (decaying into $^{10}$B with a lifetime of 1.387 Myr), entering the Be/B ratio, allows one to break the $K_0/L$ degeneracy and thus to constrain $L$ [57]. A careful analysis is necessary to interpret high-precision AMS-02 data, in order to minimise possible biases in the parameter determination [30]. This requires to account for the important effects of nuisance parameters in nuclear cross sections and Solar modulation and to use a covariance matrix for data uncertainties.

**Configurations for the transport coefficient.** The parameterisation of the diffusion coefficient is motivated by the analysis of B/C data [19, 28, 31, 58],

$$K(R) = \beta^\eta K_0 \left\{ 1 + \left( \frac{R_l}{R} \right)^{\frac{\delta - \delta_l}{s_l}} \right\}^{s_l} \left\{ \frac{R}{R_0 = 1\,\text{GV}} \right\}^\delta \left\{ 1 + \left( \frac{R}{R_h} \right)^{\frac{\delta - \delta_h}{s_h}} \right\}^{-s_h}, \tag{3}$$

where the indices $l$ and $h$ refer to some possible low- and high- rigidity breaks. We use here the three benchmark configurations from [31]: (i) BIG (parameters $K_0$, $\delta$, plus $R_l$, $\delta_l$, $s_l$, and $V_c$, $V_a$) has a double-break diffusion coefficient, convection, and reacceleration, maximising the flexibility at low rigidity[2]; SLIM (parameters $K_0$, $\delta$, and $R_l$, $\delta_l$, $s_l$) allows for a possible damping of small-scale magnetic turbulence, with a low-rigidity change of the diffusion slope without convection and reacceleration ($V_a = V_c = 0$ and $\eta = 1$); (iii) QUAINT (parameters $K_0$, $\delta$, $\eta$, and $V_c$, $V_a$) has instead a non-relativistic break of the diffusion coefficient [59, 60] in addition to convection and reacceleration.

**Best-fit values for the model parameters.** As discussed in [31], the high-rigidity break parameters have a minor impact on the other transport parameters, so that the latter are extracted from secondary-to-primary ratios, while the former are mostly constrained from primary fluxes. The transport parameter and halo size values used in this study, for the three configurations BIG, SLIM, and QUAINT, are taken from [33]. Finally, following the procedure detailed in App. II.B of [36], the high-rigidity break parameters and source slope parameters (of primary species) are determined from the fit on H, He, C, and O data (heavier species add no further constraints). At slight variance with the latter paper, where the halo size $L$ was fixed, we find[3] a small dependence of these parameters with $L$. However, we checked that using the best-fit high-rigidity break parameters at $L = L_\text{best}^\text{config}$ (best halo size derived in [33] for the respective configurations BIG, SLIM, and QUAINT) instead of using them at the value of $L$ under scrutiny leads to a difference noticeable only above 100 GeV and always much smaller than the data uncertainties.

**Calculation of secondary and primary $\bar{p}$, and propagation of uncertainties.** Equipped with the best-fit parameters, it is straightforward to compute (see App. A.2) the primary $\bar{p}$ fluxes for specified DM particle physics properties (see Eq. (13)), as well as the secondary fluxes; these calculations are performed with the USINE code[4] [61]. However *straightforward* does not mean *optimal* in terms of strategy and computational time for our purpose.

Indeed, although the calculation of a single model configuration is already quite fast (typically less than or at most a few seconds depending on the number of Bessel functions used),

---

[2]In BIG, we set $\eta = 1$ to avoid possible degeneracies with the low-rigidity break parameters.

[3]The anti-proton analysis in [36] was based on transport parameters from AMS-02 B/C data [31] whereas we rely here on the more up-to-date analysis of Li/C, Be/B, and B/C data [33]; hence the necessary update of high-rigidity break and source parameters.

[4]https://lpsc.in2p3.fr/usine

the constraints set on the primary flux parameters need several thousands of calls. While this would still be manageable, we rely on tabulated secondary and primary fluxes (calculated only once) to perform many minimisations, whose results are obtained in at most a minute. This allows us to quickly perform many tests, besides caring for the planet.

As primary $\bar{p}$ are produced all over the diffusive halo, the main 'propagation' parameter driving the DM limits is its size $L$ [35]. We thus tabulate the following quantities:

- Secondary interstellar (IS) $\bar{p}$: we calculate fluxes on a grid of ten $L$ values (in log-space) from 1 kpc to 12 kpc (values that roughly correspond to a $\sim 3\sigma$ range on $L$ values, see [33]). For each $L$, the other transport parameters are taken from the scaling relation

$$\text{parameter } \lambda_i(L) = A_i \left( \frac{L}{5\,\text{kpc}} \right)^{B_i}, \tag{4}$$

  provided in App. A of [33]. We stress that we perform this calculation in the 1D version of the model. As argued in App. A.2 of [35], the calculation in the 2D model (using the same transport parameters) can lie a few percent above or below that of the 1D model calculation, depending on the exact profile chosen for the spatial distribution of CR sources and the exact position of the radial boundary $R$. The advantage of using the 1D version is that it is faster, and that we can directly use the covariance matrix of uncertainties on secondary $\bar{p}$ calculated in [36] (see below). The only disadvantage is that we do not propagate the uncertainty from the source spatial distribution, which is anyway negligible compared to the other uncertainties (transport, source term, and cross sections).

- Primary IS $\bar{p}$: we calculate these fluxes in the 2D model on a grid of ten $L$ values in log-space from 1 kpc to 12 kpc (as for the secondary flux) and a grid of 31 DM masses (in log-space) from 7 GeV to 100 TeV. We repeat this calculation for the 3 transport configurations (BIG, SLIM, and QUAINT), for the 5 final states discussed in Sec. 2.1, and for the 3 benchmark DM profiles discussed in Sec. 2.2. At any given energy, the $\bar{p}$ flux is obtained from a log-log interpolation from the closest $L$ and $m_{\text{DM}}$ values. All these calculations are performed at a fixed $\langle \sigma v \rangle$ value, because the $\bar{p}$ flux for any other value of the annihilation cross section is simply obtained by a rescaling, which we do later at the stage of minimization.

The chosen sampling and log-log interpolations in 1D and 2D ensure a good precision on the calculated fluxes at all energies, i.e. numerical errors are well below the experimental and theoretical ones.

To compare to the data, we finally convert the above IS flux into the top-of-atmosphere (TOA) flux using the force-field approximation [62,63], which allows one to account for leading solar modulation effects [5]. The modulation level appropriate for AMS-02 data is inferred from neutron monitor data [66] and retrieved from the CR database[6] [61,67]. While charge-sign dependence effects may lead to different modulation levels for negatively charged particles [68], assuming the same modulation for antiprotons and heavier nuclei gives a satisfactory description of the data at low rigidity [36]. We avoid thus to enlarge the parameter space further.

---

[5]Note that this does not capture the whole richness of the modulation physics, for some recent works see for instance [64,65].

[6]https://lpsc.in2p3.fr/crdb, see 'Solar modulation' tab.

# 4 Statistical analysis

**Defining the likelihood function**    As traditionally in the field, we rely on the likelihood ratio *LR*:

$$LR(\mu_0) = -2\ln \frac{\sup_{\lambda \in \Lambda} \mathcal{L}(\lambda, \mu_0)}{\sup_{\{\lambda, \mu\} \in \Lambda \cup M} \mathcal{L}(\lambda, \mu)} \ . \tag{5}$$

In Eq. (5), $\lambda$ represents CR-specific parameters in their space $\Lambda$, and $\mu$ the DM-specific parameters (notably $\langle \sigma v \rangle$) in their space $M$. Contour regions at the desired $(1 - \alpha)$ C.L. can be obtained by sectioning $LR(\mu_0)$ at the height that leaves out a fraction $\alpha$ of its integral over $\mu_0$. However, defining the likelihood function $\mathcal{L}(\lambda, \mu)$ requires some care: ay there's the rub! In principle, we could define it through a 'global' $\chi^2$, measuring the distance of both nuclear and antiproton data to theory, hence:

$$-2\ln \mathcal{L}(\lambda, \mu) \equiv \chi^2_{\text{LiBeB}}(\lambda) + \chi^2_{\bar{p}}(\lambda, \mu). \tag{6}$$

As already mentioned in Sec. 3, the minimization of this sum over both CR and DM parameters would be extremely CPU-time and resources consuming. We wish to simplify and speed-up the calculation of the likelihood without losing the crucial information derived from nuclear data.

To commence, we remark that antiprotons are in large majority secondaries produced by the spallation of high-energy CR primary nuclei on Galactic gas. That component behaves like the secondary nuclear species Li, Be, B (LiBeB). We hence expect that the CR parameters $\hat{L}$ and $\hat{\lambda}_i$ minimizing $\chi^2_{\text{LiBeB}}$ should also minimize $\chi^2_{\bar{p}}$. Indeed, in [36] we have shown that using those CR parameters to calculate the antiproton flux leads to a *prediction* which is in excellent agreement with the data. In turn, in the present work we have also checked that the CR parameters best-fitting antiprotons are very close to $\{\hat{L}, \hat{\lambda}_i\}$. In the scheme BIG of CR propagation for instance, the LiBeB analysis yields $\hat{L} = 4.96$ kpc [33] to be compared to a best-fit value of 5.00 kpc assuming pure secondary antiprotons.

We also notice that the admixture of primary antiprotons produced by DM annihilations in the total flux $\Phi_{\bar{p}}$ is sub-dominant. Setting the annihilation cross section $\langle \sigma v \rangle$ at its 95% C.L. upper limit value yields a primary flux small compared to the secondary component. Hence, a fit to the data incorporating both CR and DM parameters should still yield best-fit values in the ballpark of $\{\hat{L}, \hat{\lambda}_i\}$. For the BIG scheme, a NFW halo and the $b\bar{b}$ channel, antiprotons are best-fitted for a halo height $L^*$ of 4.38 kpc, not far from $\hat{L}$ either.

Finally, rather than performing a new global analysis of propagation and DM parameters, our goal is to rely on our previous propagation studies (the latest ones being [33, 35]) to handle the evaluation of the likelihood in a reasonable computational time. To that purpose, we exploit the fact that the uncertainty on the primary antiproton flux is dominated by the uncertainty on the size of the diffusive halo, $L$, which we model as a log-normal distribution $\mathcal{N}(\log L)$ with parameters according to Table 3 (first half) of ref [33]. In Eq. (6), we essentially replace $\chi^2_{\text{LiBeB}}$ by the *posterior* probability distribution function of CR parameters derived in [32,33], where the height $L$ alone is set free. All other propagation parameters $\lambda_i$ are taken at their values $\lambda_i(L)$ best-fitting LiBeB data for a given $L$ – see Eq. (4). For a quantitative idea, the DM flux scales almost linearly with $L$, uncertain by a factor 2 [35], to be compared with the errors at most at the ten percent level due to other propagation parameters. Since optimal propagation parameters correlate with $L$, all the parameters $\lambda_i$ entering the antiproton flux $\Phi^{\text{th}}(\lambda, \mu)$ are set at their LiBeB maximum likelihood values corresponding to the considered value of $L$ (i.e. are profiled over)[7].

---

[7]In passing, the strategy followed here is the one recommended in [35], i.e. we consider the full covariance matrix of the model parameters for our hypothesis testing. Alternatively, the so-called MIN/MED/MAX benchmarks

Equipped with these notations, the likelihood function simplifies into (implicit summation over repeated indices)

$$-2\ln\mathcal{L}(\lambda,\mu) \equiv -2\ln\mathcal{L}(L,\mu) = \left\{\frac{\log L - \log\hat{L}}{\sigma_{\log L}}\right\}^2 + x_i(\mathcal{C}^{-1})_{ij}x_j. \tag{7}$$

The errors on $L$ in Table 3 (first half) of ref [33] are symmetrized to yield $\sigma_{\log L} = 0.197$. The inverse $\mathcal{C}^{-1}$ of the total antiproton covariance matrix is used to calculate $\chi^2_{\bar{p}}$. The total covariance matrix $\mathcal{C}$ includes both experimental $\mathcal{C}^{\text{data}}$ and theoretical $\mathcal{C}^{\text{model}}$ contributions, as explained in ref [36]. In each rigidity bin $i$, the flux residual is $x_i \equiv \Phi_i^{\text{exp}} - \Phi_i^{\text{th}}(L,\mu)$. The theoretical prediction $\Phi_i^{\text{th}}$ is the sum of a dominant secondary component and a primary contribution from DM.

**Reminder on $\mathcal{C}^{\text{data}}$ and $\mathcal{C}^{\text{model}}$.** The covariance matrices for the data and model have been discussed at length in [36]. For completeness and because they have a crucial impact on the analysis, we briefly recall below how the latter were calculated and some of their salient features.

The data covariance matrix, $\mathcal{C}^{\text{data}}$, was not provided by the AMS-02 collaboration. Its best estimate was built assuming $C_{ij}^\alpha = \sigma_i^\alpha \sigma_j^\alpha \exp\left(-0.5\log(R_i/R_j)^2/l_\alpha^2\right)$, with $\sigma_i^\alpha$ the relative uncertainty at rigidity bin $R_i$ and $l_\alpha$ the correlation length in unit of decade of rigidity (for the systematics $\alpha$). Taking advantage of the description of the data systematic uncertainties in the AMS-02 publication [24], we further broke down some systematics owing to their different origin, providing an educated guess for their associated $l_\alpha$ values (see also [69]). Different correlations were observed at different rigidities: at $R \lesssim 2$ GV, statistical uncertainties dominate, closely followed by the so-called 'rigidity cut-off' systematics (for which $l = 1.0$); in the intermediate regime, where a putative DM signal has been advocated, the 'acceptance' systematic uncertainty dominates ($l = 0.1$) followed by the 'XS' systematics ($l = 1.0$); finally, at $R \gtrsim 40$ GV, statistical uncertainties also dominate followed by the so-called 'Template' systematics ($l = 1.0$); we stress that although some of the systematics are fully correlated (for instance $l_{\text{scale}} = \infty$), they correspond to subdominant ones at all rigidities. As a result, the data covariance matrix of total uncertainties (including statistical and systematic uncertainties) corresponds to rigidity bins correlated on relatively small scales ($l \approx 0.1 - 1$ decade), and even independent bins at very low and very high rigidity (see Fig. 7 in the Supplemental Material of [36]).

The model covariance matrix, $\mathcal{C}^{\text{model}}$, was calculated by propagating the various sources of uncertainties entering the $\bar{p}$ flux calculation, i.e. (i) correlations and uncertainties from the transport and Solar modulation parameters (denoted 'Transport' in [36]), (ii) nuclear production cross-section uncertainties and parameter correlations (denoted 'XS'), and (iii) uncertainties from the CR flux progenitors (denoted 'Parents'), originating from the CR data flux uncertainties and associated covariance matrix of uncertainties. The 'XS' and 'Transport' uncertainties were found to be the leading ones, with 'Parent' only significant at $\gtrsim 100$ GeV/n. These three ingredients have different energy correlations, leading to a non-trivial covariance matrix of total uncertainties, with significant correlations at scales $l \gtrsim 1$ (see Fig. 8 in the Supplemental Material of [36]). In that respect, the total correlation matrix for the model is very different from that of the data.

**Setting upper limits on the DM annihilation cross section.** In order to derive bounds on $\langle\sigma v\rangle$, we rely on *Wilks' theorem*, telling us that *LR* — which is a difference of functions

---

presented in [35] could have been used for a quicker assessment of the DM constraints. We recall that these benchmarks provide a reasonable assessment of the lower/median/upper DM-induced antiproton flux (at $\approx 2\sigma$ level for the lower and upper bounds).

constructed according to Eq. (7) — is distributed as a $\chi^2_\nu$ with $\nu$ = number of degrees of freedom (dof) = dim($M$) (hence the notation $\Delta\chi^2$ is sometimes used for $LR$). Following the standard convention in the literature, we set $\alpha = 0.05$ and deduce 95% C.L. constraints for the single parameter $\langle\sigma v\rangle$ at *fixed mass and channel*, thus requiring that $LR$ attains the value 3.84 ($\chi^2$ with 1 dof). The likelihood ratio takes the form

$$LR(\langle\sigma v\rangle) = -2\ln\mathcal{L}(L_{\min}, \langle\sigma v\rangle) + 2\ln\mathcal{L}(L', \langle\sigma v\rangle'). \tag{8}$$

At fixed annihilation channel and DM mass, we derive the halo height $L'$ and cross section $\langle\sigma v\rangle'$ which maximize the likelihood (7). We then increase $\langle\sigma v\rangle$ until $LR$ reaches a value of 3.84, and that gives us the upper bound on the cross section. In the previous expression, $L_{\min}$ is the value of the halo height that maximizes the likelihood for a given value of the cross section.

**Exploring the null hypothesis.** One may also want to test the presence of a DM signal against the *null hypothesis of purely secondary production*. The test statistics of Eq. (5) is then replaced by

$$LR = -2\ln\frac{\sup_{\lambda\in\Lambda}\mathcal{L}(\lambda)}{\sup_{\{\lambda,\mu\}\in\Lambda\cup M}\mathcal{L}(\lambda,\mu)}, \tag{9}$$

i.e. the numerator of Eq. (5) is now replaced by the purely secondary production term. Again, the $LR$ takes the form of a difference of functions, here the null hypothesis vs. the alternate one. At fixed annihilation channel, this boils down to:

$$LR = -2\ln\mathcal{L}(L_{\mathrm{sec}}, \langle\sigma v\rangle \equiv 0) + 2\ln\mathcal{L}(L^*, m^*, \langle\sigma v\rangle^*), \tag{10}$$

where $L_{\mathrm{sec}}$ is the value of the halo height maximizing the likelihood with purely secondary antiprotons, whereas $m^*$ and $\langle\sigma v\rangle^*$ are the DM mass and cross section maximizing the likelihood when DM is included. In that case, the best-fit halo height is $L^*$.

However, now Wilks' theorem does not apply for *two* reasons. First, the null hypothesis corresponds to the case $\langle\sigma v\rangle \to 0$, i.e. it lies on the boundary of the allowed region. This problem can be overcome by using Chernoff's theorem [70], which can be loosely stated by saying that in such a case the $p$-value is reduced by half compared to the naive estimate. Computing this so-called *local $p$*-value would be however insufficient. A second and more challenging problem is that the signal further depends on (nuisance) parameters that are not defined under the null hypothesis, in our case the mass $m_\chi$ (and in principle the discrete parameter defining the final state, over which one also implicitly scans). The *global $p$-value* (i.e. the true one) is larger than the above-mentioned *local $p$-value*, since one must take into account the so-called *trial factor* associated to the scan over the nuisance parameter(s). This issue is rather common in the particle and astroparticle literature, see e.g. [71], where the difference between local and global significance is known as *look-elsewhere effect* (LEE). Although there is a semi-analytical understanding of the LEE [72–75], an assessment of the global $p$-value typically requires evaluating the probability distribution over mock-data sets created under the null hypothesis and can be computationally expensive. It is further customary in the literature (albeit sometimes confusing, and strictly speaking unnecessary) to associate the computed $p$-values to Gaussian $Z$ scores leading to the same significance (often in a two-tailed test). Since we will find a statistically unimportant local significance, even without further calculations we can conclude that the global $p$-value is *a fortiori* higher, and thus that the data do not allow one to reject the null hypothesis of pure secondary production. This conclusion, anticipated in [36], is thus comforted by the current analysis.

Table 2: DM best-fits based on the dataset [24]; the last column refers to the *local* significance expressed in Gaussian terms using the same convention as Ref. [69], in the benchmark NFW DM halo.

| Final state | Model | $m^*$ [GeV] | $\langle\sigma v\rangle^*$ [cm$^3$/s] | $LR$ (denom) | $LR$ (num) | $LR$ | local signif. [$\sigma$] |
|:---:|:---:|:---:|:---:|:---:|:---:|:---:|:---:|
| $b\bar{b}$ | BIG | 109.3 | 1.71e-26 | 48.37 | 51.65 | 3.28 | 1.8 |
| $b\bar{b}$ | SLIM | 109.1 | 1.48e-26 | 48.77 | 51.70 | 2.93 | 1.7 |
| $b\bar{b}$ | QUAINT | 106.7 | 4.28e-27 | 45.32 | 45.53 | 0.22 | 0.5 |
| $q\bar{q}$ | BIG | 88.5 | 4.41e-27 | 50.31 | 51.65 | 1.35 | 1.2 |
| $\mu^+\mu^-$ | BIG | 155.7 | 2.65e-23 | 49.76 | 51.65 | 1.90 | 1.4 |
| $W^+W^-$ | BIG | 106.8 | 2.20e-26 | 49.24 | 51.65 | 2.41 | 1.6 |
| $hh$ | BIG | 166.7 | 3.62e-26 | 49.28 | 51.65 | 2.38 | 1.5 |

## 5 Results and discussion

### 5.1 Significance of a DM signal

Based on the data release [24], a number of articles have found evidence for DM, notably for annihilation in $b\bar{b}$ channel, as for instance [14, 15, 21]. We find instructive to inspect the robustness of these claims once a more realistic error treatment is performed. This exercise also allows us to stress the importance of this technical ingredient in getting sound assessment of potential signals.

We translate the likelihood ratio into a *local* Gaussian score using the same convention as in [69] (as well as [19, 21]). Besides easing the comparison, this is also instructive since the authors of [69] associated actual global significances to their findings. The local significances for a DM excess for the $b\bar{b}$ channel in different propagation models, as well as for different final states in the BIG model, are reported in Tab. 2 (all for the benchmark NFW halo profile choice). The largest local significance is of only $1.8\,\sigma$, attained in the $b\bar{b}$ channel, for $\langle\sigma v\rangle^* = 1.71 \times 10^{-26}$ cm$^3$/s, $m^* = 109$ GeV, and the BIG model.

The corresponding primary, secondary, and total flux, as well as experimental measurements and data-model residuals are reported in Fig. 1. Note that the best-fit values depend somewhat from the propagation setup; we find a $\sim 14\%$ variation in mass and almost a factor 3 variation in cross section if switching to the QUAINT scheme. These differences are qualitatively similar to the differences we find with the results in [69]. On the other hand, it is worth noting that the significance we find is very similar to the one of 1.8 $\sigma$ local obtained in [69] for their "Setup 2", i.e. using the GALPROP code [8] for CR propagation as in [76] but including error-correlations similarly to what we did in [36] and here. This indicates that the significance in favour of a DM hint is rather insensitive to the specific numerical framework chosen, while it crucially depends on the treatment of the errors. In Tab. 3 we show how the significance of the best fit changes when neglecting model errors (second line), neglecting experimental covariance and working under the hypothesis of diagonal experimental errors (line three), doing both of the above (fourth line), simply including statistical uncertainties for the experimental data (fifth line), or simply including statistical uncertainties for the experimental data and neglecting model errors (sixth line). Incomplete/unrefined treatments of the errors clearly lead to artificially more significant excesses and can boost a secondary minimum into an absolute one (note the low $m^*$'s for the cov/none and diag/none cases): It is a practice

---

[8]https://galprop.stanford.edu/

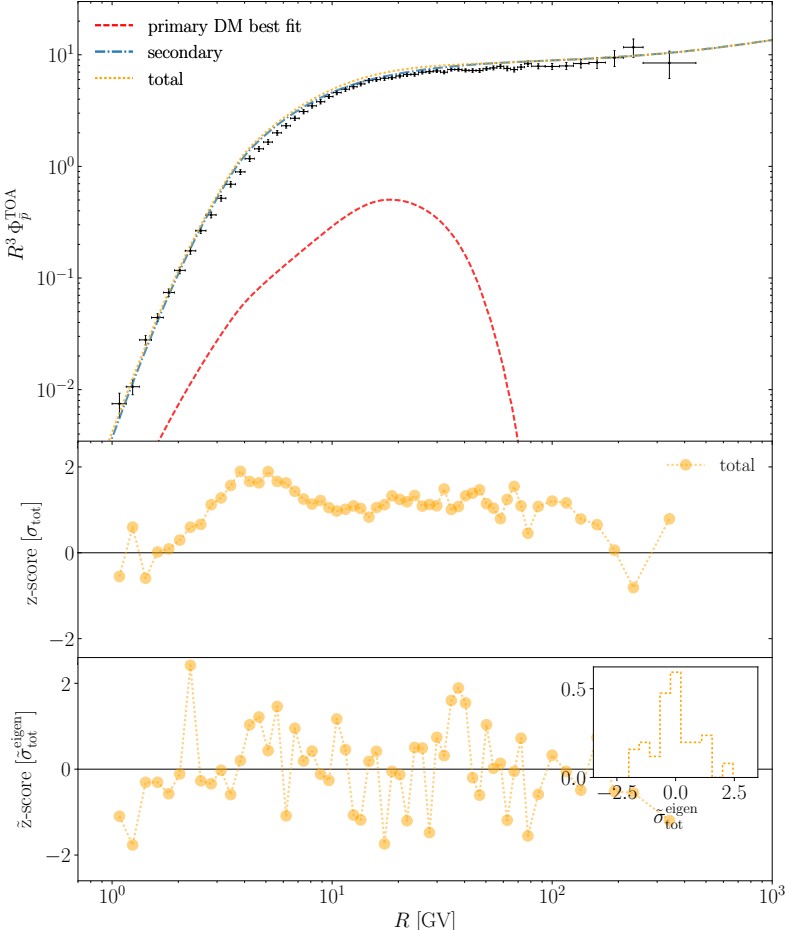

Figure 1: The secondary contribution based on the LiBeB analysis of [33] (dot-dashed blue), and the primary DM contribution in the $b\bar{b}$ channel (dashed red) in the overall best-fit to the data (dotted orange). The BIG configuration is assumed and fluxes are TOA, compared to AMS-02 data points [24]. The middle panel shows the residuals and the bottom panel the residuals in the diagonalised rigidity base, with the inset their histogram (see [36] for details).

that should be abandoned. Given that even the *local* hints for a DM signal are statistically insignificant, we do not pursue the computationally more intensive assessment of their global significance. Note, however, that this was estimated in [69] (with which we closely agree in local significance assessment) to amount to only $\sim 0.5\,\sigma$.

## 5.2 Bounds on DM

Once establishing that there is no significant hint for a DM signal, we proceed to derive the bounds on the DM annihilation cross section $\langle \sigma v \rangle$ versus its mass $m_\chi$. In Fig. 2 we report the limits for the fiducial propagation scheme BIG, the benchmark NFW galactic profile and for the five representative annihilation channels discussed in Sec. 2.1. The weakening of the bounds between 50-200 GeV for the quark, gauge boson and Higgs boson channels reflects the presence of the slight excess described above. Apart for kinematically-related thresholds, the bounds appear rather similar at the TeV scale and above, albeit slightly tighter for quark final states as opposed to gauge or Higgs bosons ones. As expected, the bound for the muon final state channel is up to 3 orders of magnitude weaker and not competitive with other

Table 3: The overall best-fit parameters and significance for BIG propagation, $b\bar{b}$ channel, and the benchmark NFW DM halo, for alternative choices of data and model errors.

| Err. data / model | local signif. $[\sigma]$ | $m^*$ [GeV] | $\langle\sigma v\rangle^*$ [cm$^3$/s] |
|---|---|---|---|
| cov/cov | 1.81 | 109.3 | 1.71e-26 |
| cov/none | 2.39 | 10.5 | 5.07e-26 |
| diag/cov | 3.33 | 98.8 | 2.14e-26 |
| diag/none | 2.75 | 8.5 | 1.70e-25 |
| stat/cov | 5.19 | 89.7 | 1.48e-26 |
| stat/none | 4.49 | 8.0 | 2.98e-25 |

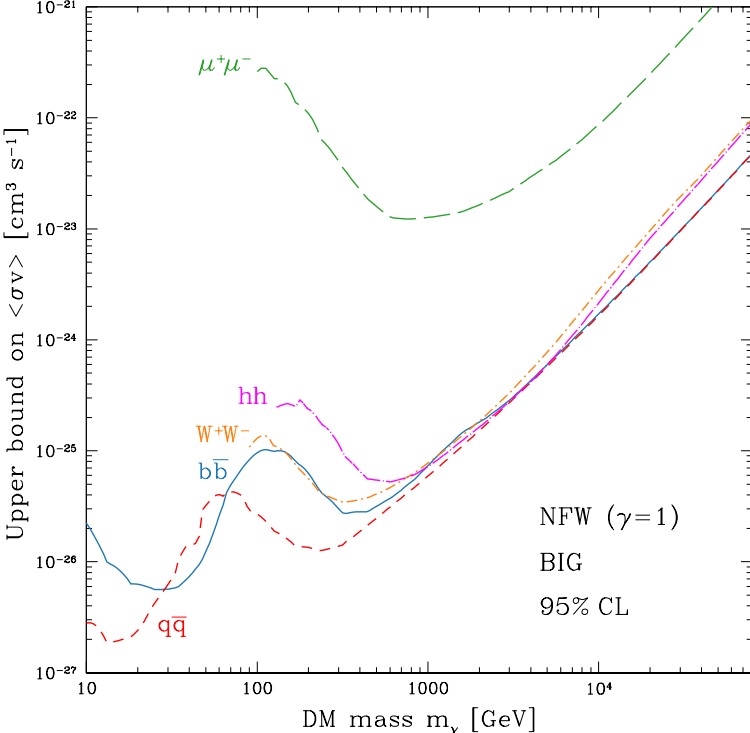

Figure 2: 95% C.L. exclusion plot for the five annihilation channels outlined in the text, the cosmic-ray propagation scheme BIG and our benchmark halo profile.

existing tighter bounds for leptonic final states, as those coming from $e^+ - e^-$ data (see for instance [22]) or cosmology [77].

In Fig. 3, left panel, we compare our limits for the benchmark case with some other results involving antiproton analyses of [24]. Compared with our previous analysis in [11], the bound is overall compatible at low masses, modulo the weakening at 50-100 GeV due to the excess (with respect to the updated model) discussed above, and the effect of marginalisation over halo thickness and modulation (absent in [11]). The much tighter bound at large masses is instead due to the fact that current propagation models, calibrated to AMS-02 secondary data, plus updated cross sections, lead to a much better agreement of secondary predictions

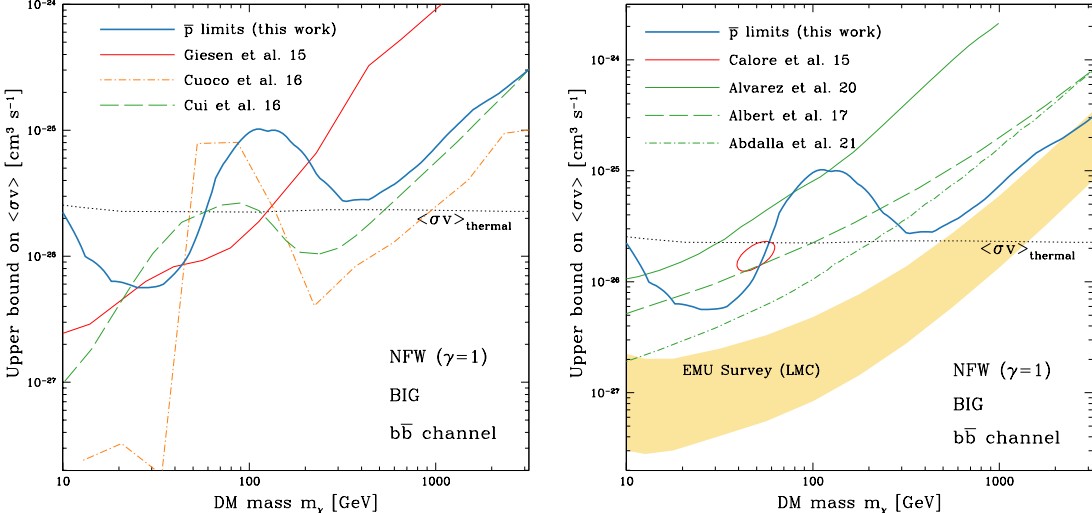

Figure 3: *Left panel:* The upper limit on the annihilation cross section derived in this work for the $b\bar{b}$ channel (blue solid line), compared to other results involving antiproton analyses of [24]. The red solid line corresponds to the Giesen et al. upper bound [11]. The limits set by Cuoco et al. [14] (orange dot-dashed curve) and Cui et al. [15] (green dashed line) are also featured. *Right panel:* The upper limit on the annihilation cross section derived in this work for the $b\bar{b}$ channel (blue solid line), compared to other probes. The red contour is the 95% CL contour of the fit to the Galactic Center excess reported by Calore et al. in [78] for the same annihilation channel. We also display bounds from different samples dwarf spheroidal galaxies (dSph) derived with a new data-driven method by Alvarez et al. in [79] (green solid line), with traditional template-fitting strategies by Albert et al. in [80] (green dashed line), and by combining Fermi-LAT and ground-based telescopes data by Abdalla et al. in [81] (Glory Duck project, green dot-dashed line). The yellow band are radio constraints derived from the EMU survey [82]. The thermal relic cross section reported in dotted black lines is the one computed in [83]. See text for details.

with $\bar{p}$ data measurements at high energy [36]. Compared with the results of [15], the bound curve agrees at high masses but is somewhat different at lower masses, which could be due to propagation model differences (c.f. our Fig. 4, right panel) and/or slightly different cross sections adopted (see Fig.3 in [15] for their impact). But the major difference with respect to their results is in the significance of the excess, which is larger in their case due to a simplified assumption of the error treatment (see discussion above). Qualitatively similar but quantitatively larger differences are found with the results of Ref. [14], with our results roughly matching the weakest ones of the ensemble of bounds shown in their Fig. 3.

The obtained bounds are subject to "theoretical systematics", notably those related to the propagation model and the halo profile. In Fig. 4, left panel, we report the bounds for the three different halo profiles reported in Tab. 1. Without loss of generality, we only show the case of $b\bar{b}$ annihilation and BIG propagation scheme, the shift for the other cases being rather similar. Since most of the antiproton signal is collected from within a few kpc [84], the role of the profile towards the Galactic center is rather mild, shifting the exclusion bound by less than a factor two [9]. Note that a comparable uncertainty is related to the normalisation of the

---

[9]Also note that, at least at the percent level, these normalisation changes would not affect the significance of

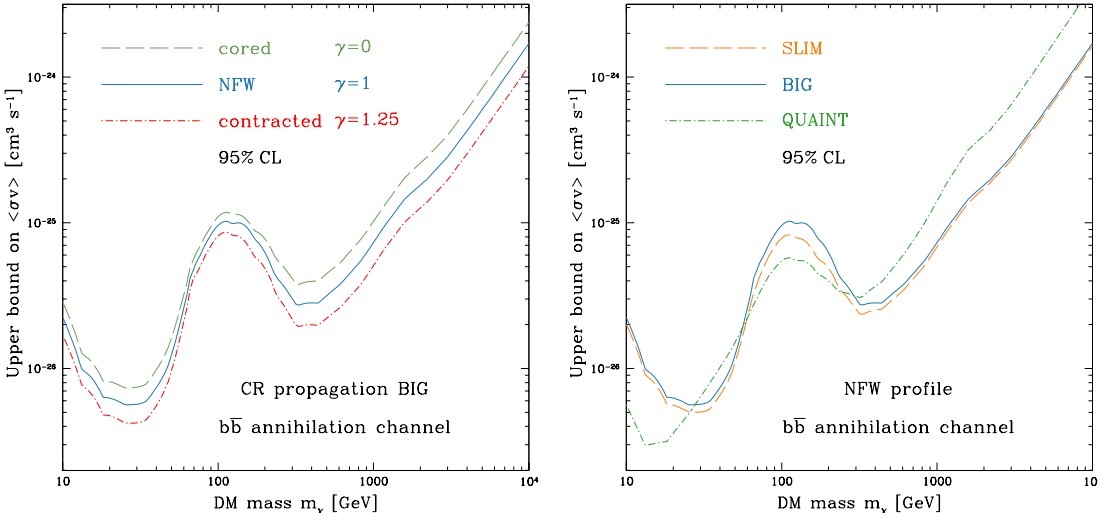

Figure 4: *Left panel:* Comparison between the limits obtained with different choices of the DM profile in the Milky Way, for the case of $b\bar{b}$ annihilation and the BIG propagation scheme. *Right panel:* Comparison between the various CR propagation schemes for our benchmark halo and the $b\bar{b}$ annihilation channel.

DM density at the solar distance from the Galactic center, amounting to about 30% in $\rho_\odot$ [85] which translates in ∼70% in the annihilation signal. A somewhat similar uncertainty is related to the propagation scheme, as illustrated in Fig. 4, right panel. Contrary to the previous case, where the uncertainty translated simply in a rescaling of the bounds, in this case the shape is also affected. The QUAINT propagation scheme leads to tighter bounds at low masses than BIG, SLIM schemes since its different functional dependence better reproduces the data at low rigidities. The opposite is true at larger rigidities.

The uncertainties just discussed are however sub-leading compared to errors sometimes introduced by an incomplete or inexact account of the data or model uncertainties. The same bias that we discussed in Sec. 5.1 in the significance of a putative DM signal also translates into the strength of bounds. We illustrate this point in Fig. 5, which reports the ratio of 95% CL bound for the "erroneous" data and/or model error treatment (same legend as in Tab. 3) with respect to the "correct" treatment. Neglecting model errors or using simply statistical errors for the data can lead up to an order of magnitude shift for the bounds. Differences in error handling are likely responsible for the largest differences in bounds that can be found in the literature. We advise the reader to carefully check the assumptions in that respect, in order to critically assess the credibility of the results.

## 5.3 Comparison with other bounds

In Fig. 3, right panel, we compare the bound obtained for the $b\bar{b}$ channel and benchmark propagation and halo model choice with multimessenger bounds in the literature, notably those coming from non-observation of an excess gamma-ray signal from an ensemble of dwarf Spheroidal galaxies (dSph) and the absence of a radio signal excess from the Large Magellanic Cloud (LMC). We also report the thermal relic cross section computed in [83] (dotted black line).

---

an excess, but only the best-fit cross section.

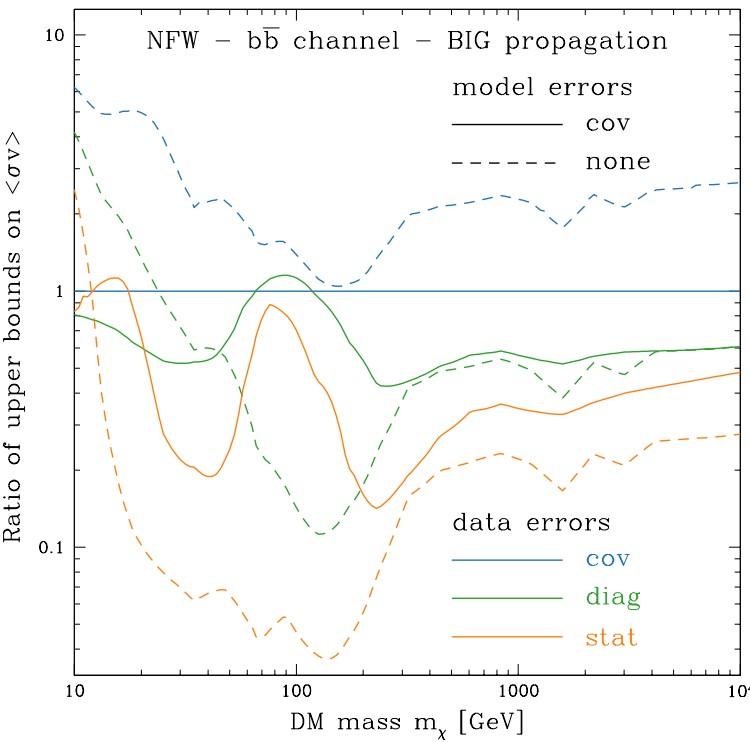

Figure 5: Comparison between limits obtained using various incarnations of data/model errors (see Table 3). Our benchmark halo is assumed with DM annihilating into $b\bar{b}$ pairs. Cosmic ray propagation is modeled within the BIG scheme.

There is no unique gamma-ray bound from dSph, since the actual constraint depends on more or less aggressive assumptions entering its derivation. We illustrate that by plotting: i) A rather conservative bound, obtained by limiting the study to four classical, kinematically well-determined dSph, and using a data-driven approach for both DM profiles and astrophysical background determination (solid green curve, from [79]; see also [86] for methodological details). ii) The bound published by the Fermi-LAT Collaboration [80], and performed with traditional template-fitting techniques of the data in a rather large region of interest around the target. This work relies on a set of 45 confirmed and candidate dwarfs, considering both classical and ultra-faint objects, but also systems with photometric properties consistent with known dSph, for which the determination of the DM distribution is largely uncertain and it is hindered by the very limited number of member stars. iii) The strongest gamma-ray bound to date, based on 20 dSph (classical and ultra-faint) and a joint likelihood analysis of Fermi-LAT and ground-based telescopes [81], dominated anyway by the Fermi-LAT sensitivity at low masses. Which bound is more "realistic" depends eventually on the degree of trust in the dSph modelling and reconstruction, which has been recognised to be a challenging aspect in particular for ultra-faint dSphs [87–91], and the astrophysical background modelling, as opposed to a pure data-driven approach.

For the LMC radio channel, we show the range of bounds inferred in [82] from a constrained value of the magnetic field and kinematical determination of the DM abundance. Note that the radio bound is coming from the synchrotron radiation that $e^{\pm}$ coming from hadronisation and decay of the $b\bar{b}$ pairs emit in the surrounding magnetic fields, which are also responsible for the $e^{\pm}$ diffusion. It is thus especially sensitive to the magnetic field deter-

mination.

The antiproton bound has a comparable strength to aggressive gamma-ray bounds at low masses, apart for the region of the small excess. The antiproton bounds are definitely better than gamma-ray ones beyond ∼300 GeV, and get comparable with the range excluded by radio data in the multi-TeV range. Given the different systematics affecting the various channels, it is worth noting how there are at least two independent channels excluding the simplest $s$−wave thermal relic DM models up to few hundreds GeV.

For illustrative purposes, we also report the 95% C.L. best-fit contour for DM explanations of the Galactic Center Excess in Fermi-LAT data at GeV energies, according to the calculation in [78]. Although we warn the reader that a direct comparison would require an ad hoc analysis (for instance, the DM profiles used in different analyses are not the same), it is clear that we now dispose of multiwavelength and multimesseger probes with adequate sensitivity to investigate the excess, whose DM interpretation is subject to tight constraints (see [22] for a dedicated analysis), despite a number of Galactic astrophysical uncertainties that invite to caution [92,93]. Similar analyses, using different propagation schemes and datasets, had been performed in the past [94–96].

## 5.4 Extended AMS-02 $\bar{p}$ dataset: a first assessment of the impact on DM

In 2021, the AMS-02 collaboration has updated its dataset for CR nuclei, $e^+e^-$, as well as antiprotons [26]. The new sample of $\bar{p}$'s, collected over 7 years, is about 60% larger than the one published in 2016 in [24] ($5.6 \times 10^5$ vs. $3.5 \times 10^5$ events). A full, consistent exploitation of this dataset for DM searches would require a number of preliminary studies on the source and propagation constraints, involving the updated datasets of CR *nuclear* species, similar to what we have been embarking on over the past 5 years [28–36], and is left for future work. Nonetheless, in the following we present a first analysis of the impact that these new data have on DM bounds, keeping in mind that the input information needed to compute the CR antiproton spectrum has not been re-calibrated to new CR nuclear data, yet. In this sense, the exercise resembles that done in [11], where the effectiveness of older models in describing new data was assessed.

We follow an analysis similar to that previously described. Note that the estimated difference in the average solar modulation potential over the concerned longer period with respect to the shorter one is below 50 MV (estimated according to Oulu neutron monitor data retrieved from https://lpsc.in2p3.fr/crdb). Since the uncertainties on the Fisk potential entering our nuisance procedure are larger, we neglect this further correction, also given the preliminary nature of this analysis. In Fig. 6, we report the bounds for the $b\bar{b}$ channel thus obtained with the 2021 dataset [24] versus the ones following from the reduced 2016 dataset [24], which we have been using in this work up to now. It is reassuring that we find similar constraints, almost overlapping at large masses while shifted by a factor ∼1.35 at low masses, leading to a maximum mismatch of the bounds of a factor ∼ 2. Overall, the best fit keeps a similar local significance, of 2.1 $\sigma$ (now attained at $m^* = 146.7$ GeV) vs. the previous 1.8 $\sigma$ (at $m^* = 109.3$ GeV). However, the $\chi^2$ obtained for both best-fit models with DM and without have degraded, from about 50 to almost 100 (with just one additional data and the same number of free parameters). This is due to two effects: i) The automatic "inflation" of $\chi^2$ (even when the absolute residuals of data minus predictions stay the same) due to a reduction of the errors, since statistical errors of the data have shrunk by about 20%, on average, and a similar reduction is also seen in the systematic errors [10]. ii) An agreement between predictions and data which is less good than the one reported in [36], mostly due to the modest

---

[10]Please note that in [26] the AMS-02 collaboration has chosen to report errors with two decimal digits in the chosen units, even when this implies that the error is reported with a single significant digit. This may sometimes imply a significant rounding error, and could contribute in a spurious way to the mentioned effect.

yet noticeable shift in the data at rigidities $\sim 1 \div 10$ GV. It is unclear at this stage if the newly acquired precision is indicating some inadequacy of the simplest propagation models to match very well the data, or if once propagation setups and parameters will have been re-adjusted to the new nuclear datasets, a significantly better agreement will be recovered, as was indeed the case after the analysis presented in [11].

Unfortunately, a comparison with the few studies that have analysed the same enlarged 2021 dataset does not fully clarify the situation. In [22], whose bounds are reported with the dashed green line in Fig. 6, the authors find significantly flattened and reduced residuals between the new data and the predictions, hence obtain very stringent bounds, systematically stronger than ours. Ref. [27] performs an updated, global reanalysis—albeit in a simplified setting with respect to the very important error treatment—limited to the assessment of the significance of the best-fit (as opposed to deriving exclusion bounds) and finds results in qualitative agreement with [22], with the evidence for DM further reduced. On the other hand, the bounds obtained in [23] are in qualitative agreement with the ones obtained here. But the focus of [23] is rather methodological, with a number of analysis techniques used, none of which exactly matches the procedure followed here. The closest one, denoted "Profiling over propagation parameters" in their fig. 10, yields the bound we reproduce in fig. 6 with a dot-dashed orange line and shows a pattern similar to ours. No comparative study between the

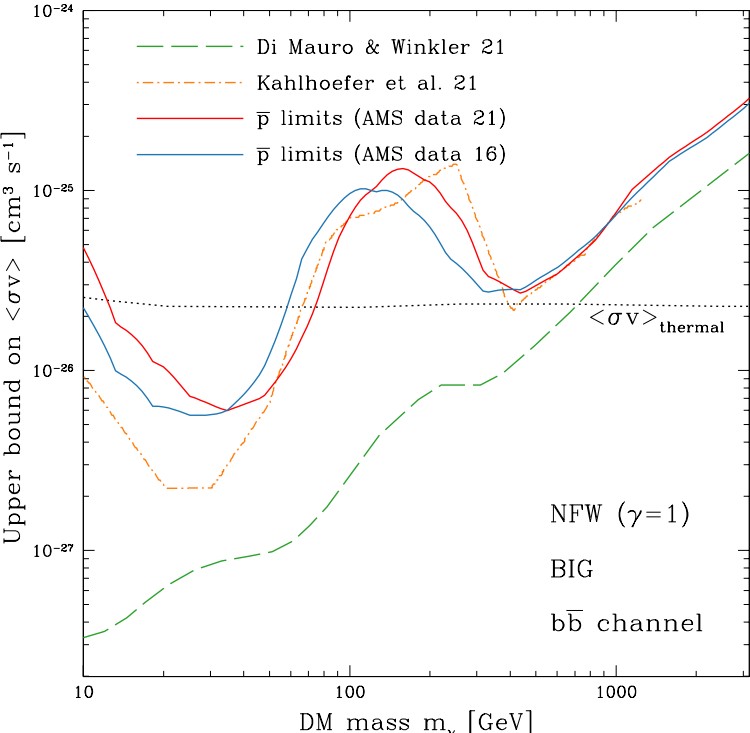

Figure 6: The upper limit on the annihilation cross section derived in this work for the $b\bar{b}$ channel based on the extended dataset in [26] (red solid line) is compared to the results recently obtained by Di Mauro and Winkler [22] (green dashed curve) and Kahlhoefer et al. [23] (orange dot-dashed line) from analysing the same data. The upper limit on the annihilation cross section derived in this work for the $b\bar{b}$ channel based on the older antiproton dataset [24] is displayed (blue solid line) for completeness.

reduced and the complete datasets is presented there to draw further conclusions. Ref. [23] also explicitly warns the reader that their "set-up is not designed to provide an accurate characterisation of this excess", because of their simplified assumptions on the error treatment and, in particular, because of the neglected error correlations.

We conclude that, while the extended 2021 AMS-02 dataset deserves further scrutiny in order to assess the viability of simplified propagation models at the current level of experimental precision, based on this preliminary study the DM bounds appear robust; their possible degradation, if any, is inferior to the overall uncertainty due to other effects previously discussed (halo profile, normalisation, propagation setup). The bounds may even tighten up, should a re-calibrated secondary prediction turn out to be in better agreement with the data.

## 6 Conclusions

In this article, we have assessed the information provided by AMS-02 antiproton data on dark matter (DM) annihilation scenarios. We have used a semi-analytical propagation model and a new calculation of DM spectra based on a recent release of the PYTHIA code to conclude that there is no significant hint for a DM excess in the data. This conclusion, which is consistent with some other results in the literature (e.g. [69]) but at odds with some older claims (e.g. [14,15, 21]), relies on a state-of-the art treatment of different sources of experimental and theoretical errors, which is the most important novel ingredient of our work. We explicitly showed how too simplistic assumptions on the errors can reflect in overestimated significances of the excesses. Moreover, we have presented bounds on the DM annihilation cross sections versus mass for a number of representative final state channels, and discussed uncertainties related to the the DM halo profile, the normalisation, and the propagation model. Finally, we have compared antiproton bounds with other bounds from gamma-ray and radio channels, finding that they are competitive with the best bounds available in the literature.

The recent AMS-02 extended data release [26] deserves further scrutiny, notably to assess its impact on the performances of existing propagation models in describing the data with sufficient precision, which we plan to embark on in the near future. They should tighten the constraints on transport parameters and primary spectra (relevant for secondaries) at high rigidity. Our preliminary analysis (section 5.4) suggests that the DM bounds presented here are rather robust, but the overall degradation of the fits, if anything, should advise for extra caution in case of new DM-like excesses were to be claimed. Note however that for most of the parameter range of interest, systematic or theory errors dominate and further statistics will not help. The most awaited output from the AMS-02 collaboration, at this stage, is a reliable assessment of the error covariance matrix, which we argued is crucial in shaping both the significance of excesses and the strength of exclusion bounds. The most promising path to reduce the theory error budget is instead related to the cross-section uncertainties, which still constitute almost half of the total uncertainty (see [36]) and can be reduced with laboratory measurements, as quantified in [97]. Fortunately, further data are expected from both LHCb and the forthcoming AMBER [11]. An improvement in the secondary prediction may additionally come from a refinement in the knowledge of light nuclei spallation cross sections (see e.g. [98] for progresses in that direction), which are partially degenerate with propagation parameters. Another promising direction would be to analyse time-binned data, if available, thus correcting more effectively for solar modulation effects.

Besides improvements in the existing channels, further advances in DM constraints or discovery potential [99–104] may come from opening up the CR *antinuclei* channels. In particular,

---

[11]https://amber.web.cern.ch

the GAPS balloon experiment, with first possible flight in late 2022[12], is expected to perform precise measurements of cosmic-ray antiprotons, anti-deuterons, and anti-helium, and complement eagerly awaited AMS-02 results with independent observations at low energies.

Finally, it would be appealing to close the gap between phenomenological models of cosmicray propagation and a more fundamental understanding of its microphysics. This is certainly one of the most challenging and stimulating frontiers to tackle in the years to come, notably if simplified models should reveal less and less capable of adequately fitting data of increasing precision.

## Acknowledgements

We thank C. Armand and M. Regis for providing us with the tabulated version of the limits presented in [81] and [82], respectively, both displayed in Fig. 3, right panel. Y.G. acknowledges support from VILLUM FONDEN under project no. 18994.

## A  Primary and secondary $\bar{p}$ flux calculation

### A.1  Transport equation for antiprotons

Cosmic-ray transport in the Galaxy can be described by a diffusion-loss equation [4, 105, 106]. The steady-state equation for the differential density of CR antiprotons per momentum unit (denoted interchangeably as $n^{\bar{p}}$ or $dn^{\bar{p}}/dp$ below) can be written

$$
\begin{aligned}
\vec{\nabla} \cdot \left(\vec{j}\, n^{\bar{p}}\right) &+ \frac{\partial}{\partial p}\left(\left(\dot{p} + \frac{p}{3}(\vec{\nabla} \cdot \vec{V}_c)\right)n^{\bar{p}}\right) - \frac{\partial}{\partial p}\left(p^2 K_{pp} \frac{\partial}{\partial p} \frac{1}{p^2} n^{\bar{p}}\right) \\
&= \dot{q}^{\bar{p}} - \sum_{t \in \mathrm{ISM}} \left(n_{\mathrm{ISM}}^t(\vec{r}) \times v \times \sigma_{\mathrm{inel}}^{\bar{p}+t}(p)\right) n^{\bar{p}}.
\end{aligned}
\tag{11}
$$

In the left-hand side, the three terms correspond to: (i) the divergence of the diffusion and convection currents, $\vec{j} = -K\vec{\nabla} + \vec{V}_c$, with $K$ the spatial diffusion coefficient and $\vec{V}_c$ the convective wind; (ii) various energy loss terms, including ionisation and Coulomb losses ($\dot{p} \equiv dp/dt$) and adiabatic losses; and (iii) reacceleration with $K_{pp}$ the diffusion coefficient in momentum space. The second term on the right-hand side corresponds to a sink term $n_{\mathrm{ISM}}^t v \sigma_{\mathrm{inel}}^{\bar{p}+t}$ related to the destruction of $\bar{p}$ on various targets of the interstellar medium (ISM), with $n_{\mathrm{ISM}}^t$ the density of the $t$-th ISM component, $v$ the $\bar{p}$ velocity, and $\sigma_{\mathrm{inel}}^{\bar{p}+t}$ the inelastic cross section on target $t$. Finally, the first term on the right-hand side is a generic source term $\dot{q}^{\bar{p}} = dq^{\bar{p}}/dt$ for the number density of $\bar{p}$ per energy and time unit ($\dot{q}^{\bar{p}}$ is short for $d\dot{q}^{\bar{p}}/dp$) that can be further decomposed into

$$
\dot{q}^{\bar{p}}(\vec{r}, p) = \dot{q}^{\mathrm{prim}}(\vec{r}, p) + \dot{q}^{\mathrm{sec}}(\vec{r}, p) + \dot{q}^{\mathrm{ter}}(\vec{r}, p).
\tag{12}
$$

These three components correspond to the so-called *primary* (from annihilating DM distributed in the DM halo), *secondary* (from standard nuclear interactions of CRs on the ISM), and *tertiary* (inelastic but non-annihilating $\bar{p}$ interactions on the ISM) contributions. They can be written

---

[12] https://gaps1.astro.ucla.edu/gaps/news.html

as

$$\dot{q}^{\text{prim}}(\vec{r}, E) = \rho_{\text{DM}}^2(\vec{r}) \times \frac{\langle \sigma v \rangle}{2 \xi m_\chi^2} \times \sum_f B_f \frac{d n^{\chi\chi \to f \to \bar{p}}}{dE}(E), \tag{13}$$

$$\dot{q}^{\text{sec}}(\vec{r}, E) = \int_{E'_{\text{th}}}^{\infty} dE' \sum_{c \in \text{CRs}} \left[ \sum_{t \in \text{ISM}} \left( n_{\text{ISM}}^t \times v' \times \frac{d\sigma_{\text{prod}}^{c+t \to \bar{p}}}{dE}(E', E) \right) \frac{dn^c}{dE'}(E') \right], \tag{14}$$

$$\dot{q}^{\text{ter}}(\vec{r}, E) = \left\{ \int_E^{\infty} dE' \left[ \sum_{t \in \text{ISM}} \left( n_{\text{ISM}}^t \times v' \times \frac{d\sigma_{\text{nar}}^{\bar{p}+t \to \bar{p}}(E', E)}{dE} \right) \frac{dn^{\bar{p}}}{dE'}(E') \right] \right\} \tag{15}$$
$$- \sum_{t \in \text{ISM}} \left( n_{\text{ISM}}^t \times v \times \sigma_{\text{ina}}^{\bar{p}+t}(E) \right) \frac{dn^{\bar{p}}}{dE}(E),$$

where we explicitly write the energy $E$ of the outgoing $\bar{p}$, and where primed quantities in the integrals, $E'$ and $v'$, correspond to the incoming energy and velocity of the CR involved in the production of $\bar{p}$. Let us review the quantities entering these source terms:

- $\dot{q}^{\text{prim}}$ involves the DM density $\rho_{\text{DM}}$ (described in Sect. 2.2), the thermally averaged annihilation cross section $\langle \sigma v \rangle$, the mass of the DM candidate $m_\chi$ (with $\xi = 1$ or 2, whether the DM particles are or are not self-conjugate), the branching ratio $B_f$ (where $f$ is for instance $b\bar{b}$, $W^+W^-$, etc.), and the production per annihilation of $\bar{p}$, $dn^{\chi\chi \to f \to \bar{p}}/dE$ (described in Sect. 2.1), via the final state $f$.

- $\dot{q}^{\text{sec}}$ involves the production of $\bar{p}$ by impinging CRs (differential density $dn^c/dE'$) on ISM targets, $n_{\text{ISM}}^t$, via the differential production cross section $d\sigma_{\text{prod}}^{c+t \to \bar{p}}/dE$; the energy threshold $E'_{\text{th}}$ to produce $\bar{p}$ is $7m_p$ (total energy). Following our previous study of secondary $\bar{p}$ [36], the calculation accounts for CR parents (indexed by $c$) going from H to Fe, and we use the cross-section parametrisation 'Param II' of Refs [97,107], along with the scaling relation 'B' of Ref. [97] for nucleon-nucleon interactions.

- $\dot{q}^{\text{tert}}$ involves the non-annihilating rescattering differential cross section $d\sigma_{\text{nar}}^{\bar{p}+t \to \bar{p}}/dE$, that is the probability for a $\bar{p}$ of energy $E'$ to survive the interaction and end up at an energy $E < E'$. This differential cross section is taken from [108] with the energy dependence from [109], and the inelastic non-annihilating total cross section $\sigma_{\text{ina}}^{\bar{p}+t} = \sigma_{\text{inel}}^{\bar{p}+t} - \sigma_{\text{nar}}^{\bar{p}+t}$ is taken from [110]. Note that the second term in Eq. (15) ensures that the net balance of $\bar{p}$ in the tertiary term is zero (neither creation nor destruction of the total number of $\bar{p}$, only an energy redistribution).

## A.2 Equation and solution for cylindrical symmetry (2D)

The solution of Eq. (11) depends on the geometry of the system, the spatial and momentum dependence of the various transport parameters, the source terms considered, and the boundary conditions. In this work, we assume a simplified 2D cylindrical geometry of radius $R = 20$ kpc, with the Sun located at a radius $R_\odot$. CRs propagate in a diffusive halo of half-thickness $L$. CRs can also be advected by a constant vertical wind originating from the disc ($\vec{V} = \pm V_c \vec{e}_z$). The gas density is assumed to be constant in a thin disc of width $2h = 200$ pc, so that energy losses and nuclear interactions only happen in this region; reacceleration is also restricted to this thin disc. If we further pinch the disc mathematically into $2h\delta(z)$ [111], we can obtain a compact form for the differential density of CR $\bar{p}$. The corresponding equation and solutions for secondary and primary source terms have been given in several previous publications [112,113], but we repeat them below for completeness.

**Transport equation in cylindrical symmetry.** In cylindrical symmetry with the above assumptions (and expressed as a function of total energy $E$), Eq. (11) becomes for $n^{\bar{p}}(r, z, E)$

$$- \left[ K \left( \frac{\partial^2}{\partial z^2} + \frac{1}{r} \frac{\partial}{\partial r} \left( r \frac{\partial}{\partial r} \right) \right) - V_c \frac{\partial}{\partial z} \right] n^{\bar{p}} + 2h \, \delta(z) \frac{\partial}{\partial E} \left[ B(E) \, n^{\bar{p}} - C(E) \frac{\partial n^{\bar{p}}}{\partial E} \right]$$
$$= \dot{q}^{\,\mathrm{prim}}(r, z, E) + 2h \delta(z) \left[ \dot{q}^{\,\mathrm{sec}}(r, z, E) + \dot{q}^{\,\mathrm{ter}}(r, z, E) \right] - 2h \, \delta(z) \Gamma^{\bar{p}}_{\mathrm{inel}} n^{\bar{p}}(r, z, E), \quad (16)$$

with $r$ and $z$ the radial and vertical coordinates ($z = 0$ is the thin disc) and the coefficients $B(E)$ and $C(E)$ given by[13]

$$B(E) = \left\langle \frac{dE}{dt} \right\rangle_{\mathrm{ion,Coul.}} + \left\langle \frac{dE}{dt} \right\rangle_{\mathrm{Adiab.}} + \left\langle \frac{dE}{dt} \right\rangle_{\mathrm{Reacc.}} \quad (17)$$
$$= \left\langle \frac{dE}{dt} \right\rangle_{\mathrm{ion,Coul.}} - E_k \left( \frac{2m + E_k}{m + E_k} \right) \frac{V_c}{3h} + (1 + \beta^2) \frac{K_{pp}}{E},$$
$$C(E) = \beta^2 \times K_{pp}. \quad (18)$$

In the above equations, formulae for ionisation losses on neutral matter and Coulomb losses in the ionised ISM are taken from [114, 115], and we also wrote the nuclear interaction term in the thin disc as a rate, i.e.

$$\Gamma^{\bar{p}}_{\mathrm{inel}}(E) = \sum_{t \in \mathrm{ISM}} \left( n^t_{\mathrm{ISM}} \times v \times \sigma^{\bar{p}+t}_{\mathrm{inel}}(E) \right). \quad (19)$$

For definiteness, we take $n_{\mathrm{ISM}} = 1 \, \mathrm{cm}^{-3}$ (with 90% H and 10% He in number), and $\langle n_e \rangle = 0.033 \, \mathrm{cm}^{-3}$ and $T_e = 10^4$ K [116], and we follow Refs. [29, 32] and Refs. [117–119] respectively for the diffusion coefficient $K(R)$ (provided in the main text, see Eq. 3) and the diffusion coefficient in momentum

$$K_{pp} = \frac{4}{3} V_a^2 \beta^2 E^2 \frac{1}{\delta(4 - \delta^2)(4 - \delta)K(R)}. \quad (20)$$

The transport parameters of the model are related to the parameters in the diffusion coefficient (slope $\delta$, normalisation $K_0$, and possible low- and high-rigidity breaks), $V_a$ which mediate the reacceleration in $K_{pp}$, and the convective wind $V_c$: the values used for this analysis are discussed in the main text (Sect. 3).

**Solution in cylindrical symmetry.** To solve Eq. (16), we rely on a Fourier-Bessel expansion along the radial coordinate $r$ of a function $f(r, z, E)$:

$$f(r, z, E) = \sum_{i=1}^{\infty} f_i(z, E) J_0 \left( \zeta_i \frac{r}{R} \right), \quad (21)$$

$$f_i(z, E) = \frac{2}{R^2 J_1^2(\zeta_i)} \times \int_0^R r \, f(r, z, E) J_0 \left( \zeta_i \frac{r}{R} \right) dr, \quad (22)$$

where $J_0$ and $J_1$ are Bessel functions of order 0 and 1 respectively, $f_i(z, E)$ are the Fourier-Bessel coefficients, and $\zeta_i$ the zeroes of $J_0$, i.e. $J_0(\zeta_i) = 0$. Applied on $n^{\bar{p}}(r, z, E)$, this expansion automatically ensures the boundary condition $n^{\bar{p}}(r = R, z, E) = 0$.

Using the property $\nabla_r J_0(\zeta_i r/R) = -(\zeta_i/R)^2 J_0(\zeta_i r/R)$, Eq. (16) turns into differential equations for the Fourier-Bessel coefficients $n_i(z, E)$ (for simplicity, we omit the $\bar{p}$ superscript

---

[13] The first order term for reacceleration in $B(E)$ comes from the expansion of the third parenthesis of Eq. (11).

in the following):

$$- K\left(\frac{\partial^2}{\partial z^2} - V_c\frac{\partial}{\partial z} + \frac{\zeta_i^2}{R^2}\right) n_i(z,E) + 2h\,\delta(z)\frac{\partial}{\partial E}\left(B(E)\,n_i(z,E) - C(E)\,\frac{\partial n_i}{\partial E}(z,E)\right)$$
$$= \dot{q}_i^{\text{prim}}(z,E) + 2h\delta(z)\left(\dot{q}_i^{\text{sec}}(z,E) + \dot{q}_i^{\text{ter}}(z,E)\right) - 2h\,\delta(z)\Gamma_{\text{inel}}n_i(z,E), \tag{23}$$

where $\dot{q}_i^{\text{prim}}(z,E)$, $\dot{q}_i^{\text{sec}}(E)$, and $\dot{q}_i^{\text{ter}}(E)$ are Fourier-Bessel coefficients of the source terms. In the diffusive halo, energy gains and losses and nuclear interactions vanish, and only the primary source term (DM annihilations) remains. The solution (w.r.t. the $z$ coordinate) is even, and with the boundary conditions $n_i(z=\pm L,E)=0$ and ensuring the continuity through the thin disc, we get (in the upper-half region) [113][14]

$$n_i(z,E) = n_i^{\text{src disc}}(0) \times \exp\left(\frac{V_c z}{2K}\right)\frac{\sinh\left(S_i(L-z)/2\right)}{\sinh(S_i L/2)}$$
$$+ n_i^{\text{src halo}}(0) \times \exp\left(\frac{V_c z}{2K}\right)\left[\cosh\left(\frac{S_i z}{2}\right) + \frac{(V_c + 2h\Gamma_{\text{inel}})}{KS_i}\sinh\left(\frac{S_i z}{2}\right)\right] - \frac{y_i(z)}{KS_i}, \tag{24}$$

with

$$y_i(z) = 2\int_0^z \exp\left(\frac{V_c(z-z')}{2K}\right) \times \sinh(S_i(z-z')/2) \times \dot{q}_i^{\text{prim}}(z')dz', \tag{25}$$

$$A_i = V_c + 2h\Gamma_{\text{inel}} + KS_i\coth\left(\frac{S_i L}{2}\right), \tag{26}$$

$$S_i = \sqrt{\left(\frac{V_c}{K}\right)^2 + \left(\frac{2\zeta_i}{R}\right)^2}. \tag{27}$$

Without energy losses and gains, we get

$$\hat{n}_i^{\text{src disc}}(z=0,E) = \frac{2h\dot{q}_i^{\text{sec}}}{A_i}, \tag{28}$$

$$\hat{n}_i^{\text{src halo}}(z=0,E) = \exp\left(\frac{-V_c L}{2K}\right)\frac{y_i(L)}{A_i\sinh(S_i L/2)}, \tag{29}$$

but the full solution is obtained by solving[15] (with 'src' either in the disc or the halo[16])

$$n_i^{\text{src}}(0) + \frac{2h}{A_i} \times \frac{d}{dE}\left(Bn^{\text{src}}(0) - C\frac{dn_i^{\text{src}}}{dE}(0)\right) = \hat{n}_i^{\text{src}}(0). \tag{30}$$

This equation is solved using a finite difference scheme (with boundary conditions) and amounts to a tridiagonal inversion [112]. A detailed description of the boundary conditions as well as the stability of the numerical scheme is provided in App. C and D of [30].

Once we get $n_i^{\bar{p}}(0) = n_i^{\text{src disc}}(0) + n_i^{\text{src halo}}(0)$, we are interested in the flux in the Solar neighbourhood ($r = R_\odot$), given by

$$n_\odot^{\bar{p}}(E) = \sum_{i=1}^\infty n_i^{\bar{p}}(0)J_0\left(\zeta_i\frac{R_\odot}{R}\right). \tag{31}$$

---

[14]Note that there is a misprint in Eq. (A5) of [113], where $KA_i S_i$ should be replaced by $KS_i$.

[15]The tertiary source term is not included in Eq. (30), because it would formally correspond to an integro-differential equation on $n_i^{\bar{p}}$. The trick is to extract the solution without the tertiary terms, i.e. a source term $\hat{n}_i^0(0) = \hat{n}_i^{\text{src disc}}(0)$, then to update the solution iteratively from the update source term $\hat{n}_i^{j+1}(0) = \hat{n}_i^j(0) + 2h\dot{q}_i^{\text{tert}}/A_i$. In practice, a couple of iterations are enough to converge to the desired solution.

[16]DM sources outside the diffusive halo are negligible, see App. B of [113].

This differential density is then converted into a differential flux as a function of the kinetic energy $E_k$ (we assume isotropy),

$$J_{\text{IS}}^{\bar{p}}(E_k) \equiv \frac{dJ_{\text{IS}}^{\bar{p}}}{dE_k}(E_k) = \frac{v}{4\pi} \times n_{\odot}^{\bar{p}}(E_k) \,. \tag{32}$$

**Primary contribution with** USINE. The above 2D solution is implemented in the USINE semi-analytical propagation code [120]. This 2D geometry has been successfully used in many previous studies of DM-related CR signals [10, 35, 56, 121] and also, in its 1D version, to describe recent AMS-02 nuclear data [32, 33] and secondary $\bar{p}$ [36] (also with USINE).

For the primary contribution, all our calculations rely on USINE v4.0 (in prep.), which includes some improvements and checks for the precision of primary $\bar{p}$ calculations. In particular, a complication occurs because the DM distribution in the MW strongly varies on very small scales near the Galactic centre: to describe correctly a variation at the scale $\lambda_n$, at least $n \geq \pi R/(2\lambda_n)$ Bessel orders (in the Fourier-Bessel expansion) are necessary. This translates into $3 \times 10^4$ Bessel functions, in principle, to describe sources varying at the pc scales in the Galactic centre (which would be prohibitive and unstable in terms of computation time). However, far away from the observer, the propagator is smoothly varying, so that at $R_{\odot}$ (where we calculate the flux), all the DM in the Galactic centre can be assimilated as a point source. The trick is to replace the initial source function by a smoother one below $r_0$ (enforcing continuity of the function and its derivative at $r_0$), conserving the number of DM-induced particles. The scale $r_0$ must be large enough, so that few Bessel functions are needed in the expansion, but not too large, so that $r_0 \ll R_{\odot}$. Different possibilities have been discussed [122, 123], and we use here the form proposed in [123] to replace the DM spherical distribution $q_{\text{DM}}(r)$ by its smoothed counterpart $q_{\text{DM}}^{\star}(r)$:

$$q_{\text{DM}}^{\star}(r) = \begin{cases} q_0 \left( 1 + a_1 \text{sinc}\left(\frac{\pi r}{r_0}\right) + a_2 \text{sinc}\left(\frac{2\pi r}{r_0}\right) \right) & \text{for } r \leq r_0 \\ q_{\text{DM}}(r) & \text{otherwise.} \end{cases} \tag{33}$$

For a DM profile $q_{\text{DM}}(r) \propto r^{\gamma}$, we get [123]

$$a_1 = a_2 + 2\gamma, \tag{34}$$

$$a_2 = \frac{8\gamma(\pi^2 - 9 + 6\gamma)}{9(3 - 2\gamma)}. \tag{35}$$

In practice, we find that using $r_0 = 500$ pc with $N = 20$ Bessel functions (using the resummation coefficients proposed in [124] to accelerate the convergence of the Fourier-Bessel series) ensures a numerical error on the primary $\bar{p}$ fluxes computed at Earth below the percent level.

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
