# Peer review of "AMS-02 antiprotons and dark matter: Trimmed hints and robust bounds"

_SciPost Physics, doi:SciPost Phys. 12, 163 (2022)_

## Round 1 · Referee Report · Anonymous (Referee 1) · 2022-2-19

Strengths

1. Excellent evaluation of the way the treatment of errors impacts the significance of a dark matter signal.
2. Sensible discussion of robust procedures for assessing local significance, with cautionary words about simplistic error analyses.
3. Up-to-date cosmic ray propagation models used.
4. Useful and reliable indirect detection limits on the dark matter annihilation cross section are calculated and presented in the plots.

Report

This manuscript makes a very useful addition to the literature on dark matter indirect detection signals. The analysis is robust. The results are interesting and relevant. I have no hesitation in recommending publication.

  • validity: top
  • significance: high
  • originality: high
  • clarity: top
  • formatting: excellent
  • grammar: perfect

Author:  Marco Cirelli  on 2022-03-31  [id 2344]

(in reply to Report 1 on 2022-02-19)

Dear Referee,
thank you very much for your positive assessment. We are submitting a slightly revised version in response to comments from Referee#2. We attach a PDF file with the modifications highlighted in color.
Best regards, the authors

Attachment:

antip_analysis_wnASloX.pdf

---

## Round 1 · Referee Report · Anonymous (Referee 2) · 2022-3-8

Strengths

1- Detailed discussion of analysis strategy and clear presentation of results
2- Careful treatment of experimental and theoretical uncertainties
3- Methodology sets the standard for any future analysis of AMS-02 anti-proton data

Weaknesses

1- Discussion of results from the literature could be more pedagogical
2- Not all approximations are sufficiently justified, analysis might be overly conservative
3- Most results not new, considered data set somewhat outdated

Report

In the manuscript "AMS-02 \bar{p}’s and dark matter: Trimmed hints and robust bounds" the authors weigh in on the important discussion about the interpretation of the AMS-02 anti-proton data in the context of dark matter annihilation. Over the past years, there have been various claims regarding the evidence of a dark matter signal in this data set (and its significance) and how the data can be used to constrain dark matter models.
In this context, the present work emphasizes the need of an accurate treatment of experimental and theoretical uncertainties, which substantially reduce the significance of the dark matter excess and allow robust bounds to be obtained (there’s the respect!). At the same time, the authors emphasize the need for efficient analysis strategies that do not waste time and resources (although it does not always become clear how much more computationally expensive a more detailed analysis would be, see also below).
I find the paper extremely well written and very enjoyable to read. The authors are clearly world-leading experts on the topic, and their experience and authority is highly welcome. I very strongly agree with the central point of this paper, which is that an accurate treatment of uncertainties (and their correlations) is essential. Although no substantial new results are presented and the data set under consideration is already outdated, it is very important for the community to have such a careful and detailed analysis. I expect the paper to be suitable for publication in SciPost once the authors have clarified whether their treatment of uncertainties is indeed accurate rather than conservative. Specifically, I would ask them to address the following comments.

Requested changes

Major comments:
1- The authors have written many previous publications on related topics and sometimes seem to assume that the reader is familiar with most of them. I find this most striking in the context of the covariance matrix for theoretical uncertainties, for which no details are provided and the reader is simply referred to ref. [36]. I think it would be important to briefly review the key elements of this approach and its justification (and make an effort to reduce the overall number of places where the reader is forced to consult another paper to understand the analysis).
2- The authors take a significant shortcut in their analysis by assuming that theoretical uncertainties can be represented by a covariance matrix that can be added to the one describing correlations in experimental data. Doing so is clearly conservative, in the sense that error bars increase and hence the preference for (or evidence against) any signal hypothesis is reduced. A more realistic treatment should instead associate the theoretical uncertainties with the signal prediction, for example by introducing nuisance parameters that are profiled out at each step. I don't think such an approach needs to be computationally very expensive (see e.g. https://cds.cern.ch/record/2242860). Crucially, using nuisance parameters to encode systematic uncertainties may increase rather than decrease the significance of a local excess, if the nuisance parameters can be adjusted in such a way that they improve the fit of the excess.
3- An important aspect of the present work is that it no longer uses the MIN-MED-MAX scheme advertised by some of the authors in previous studies (see e.g. arXiv:2103.04108). It would be good to clarify whether the authors now consider this scheme inappropriate for scrutinising the AMS-02 anti-proton data and would discourage its use, or whether they expect that it would lead to largely similar results. Along similar lines, the appendix of arXiv:2103.04108 advertises the use of a covariance matrix to capture the effect of uncertainties in propagation parameters. Would it not be possible to use this approach in the present work?
4- It remains unclear why the QUAINT propagation scheme leads to weaker constraints than the BIG propagation scheme, even though the latter has more free parameters and should therefore in principle give less tight bounds. A short discussion of this should be added.

Minor comments:
1- Since the authors have chosen to use fairly mathematical language to describe their statistical approach, I would like to add that Wilks theorem does not apply if the null hypothesis lies at the boundary of the parameter space and one needs to use Chernoff's theorem instead (see https://arxiv.org/pdf/1407.6617.pdf). As a result, the critical value of Delta chi^2 corresponding to a 95% confidence level exclusion is reduced from 3.84 to 2.71.
2- The plots comparing the present work to previous studies are very important, but quite inconvenient for the reader, as the text never connects the author names (used in the plot legend) to the bibliography entry. I would suggest making this connection explicitly in the figure caption or in the main text.
3- In order to enable a better comparison with the literature in figure 6 it would be more appropriate to use the QUAINT rather than the BIG setup.
4- It is not clear why the bound for the WW final state stops at m_chi = m_W, given that annihilation can also proceed through an off-shell W boson. This bound should be extended if possible.

  • validity: high
  • significance: high
  • originality: good
  • clarity: top
  • formatting: excellent
  • grammar: perfect

Author:  Marco Cirelli  on 2022-03-31  [id 2343]

(in reply to Report 2 on 2022-03-08)

Dear Referee,
we would like to thank you for your detailed and in-depth comments. Please find our responses below. We also attach a PDF version of the modified paper where we highlight in color the modifications we implemented, to allow you to spot them more easily.
Best regards, the authors

Attachment:

antip_analysis.pdf

---

## Round 2 · Referee Report · Anonymous (Referee 1) · 2022-4-1

Report

In the revised version of the manuscript, the authors have added extensive discussion that carefully details their statistical handling of uncertainties. I recommend the manuscript as suitable for publication.
  • validity: -
  • significance: -
  • originality: -
  • clarity: -
  • formatting: -
  • grammar: -

Author:  Marco Cirelli  on 2022-04-02  [id 2348]

(in reply to Report 1 on 2022-04-01)
Category:
answer to question

Dear Referee, thank you for your positive recommendation. We are very happy that you found the new version suitable. We have the doubt though: in addition to the revised version, we had attached a detailed response to your comments in text format. We have the impression that this went lost inn the response form. We copy here below for the records, even if it is probably now useless. We thank you very much again and we apologize for the inconvenience. Best regards, the authors

Answer to major comments:

1-We agree with the referee that we probably point the reader to a lot of our previous publications instead of giving self-consistently all the details and explanations. This was done on purpose to avoid long technical discussions. However, we agree that this may be frustrating for the reader, especially where the covariance matrices for the model and data are concerned, as they are key to the analysis. For this reason, we added a paragraph in Sec. 4 recalling how they are built and their most salient features.

2- We do agree with the referee that our treatment of the uncertainties is conservative; we also agree with the referee on the fact that profiling over the parameters would not be computationally much more costly. However we think that in our situation our approach is realistic and fair for the following reasons.
The 'profiling approach' indicated by the referee would amount to pick up one model configuration, without adding to the error matrix the corresponding uncertainty. There are two problems with applying this approach, one quantitative and one conceptual. The quantitative challenge is that most configurations in model space are almost equally probable (even when parameters are sometimes rather different), so that picking one would be statistically not very meaningful while resulting in over-optimistic estimates of the antiproton observable uncertainties. For example, the uncertainty on the halo thickness $L$ is quite large (~50 percent) and this effective parameter is to large extent degenerated with $\langle \sigma v \rangle$. Merely profiling on $L$ might choose an excessively large value for $L$, resulting in tight bounds on $\langle \sigma v \rangle $ with no statistically sound meaning. As the referee points out, this would lead to more aggressive results (significance of excesses and bounds), which would be however not very robust. Since a caveat on the robustness of a number of claims concerning antiprotons is the main message of our paper, it seems consistent to us to avoid adopting this approach and opt for the method we have used . The more conceptual issue is that profiling over the parameters requires some confidence that the 'correct' model is among the ones span in the model space. But propagation models are effective (as opposed to first principles) models: Currently, most indications are that in the range they span, they can reasonably include/describe the data within current uncertainties; but their capability to include a 'perfect' model is much more questionable.
The situation is thus quite different from the one mentioned by the referee, https://cds.cern.ch/record/2242860 , in which: a) the model parameters are much more constrained; b) we are talking of a much more controlled and well-understood situation, in the context of collider and standard model physics.

3- Actually, there is no contradiction between the approach adopted here and previous publications. In the published version of the preprint mentioned by the referee, arXiv:2103.04108 (see https://journals.aps.org/prd/abstract/10.1103/PhysRevD.104.083005) we indeed encourage the use of the covariance matrices (the approach followed in the present paper) instead of benchmarks in order to perform hypotheses testing. However the MIN/MED/MAX benchmarks are still handy for a quick inspection of the antiproton constraints in a given DM model. The MIN/MED/MAX benchmarks have shown to provide a reasonable assessment of lower/upper bound of the DM-induced antiproton fluxes at 2 sigma level, so we do expect one would obtain similar results if adopting that more approximate approach. We have added a footnote in sec. 4 to clarify this point.

4- There are two aspects raised by the referee's remark: a) The referee is right in that the QUAINT propagation scheme contains less parameters than the BIG propagation scheme, however at low rigidity they are not described by the same functional forms: a broken power-law of the rigidity for BIG and a power-law of the velocity in QUAINT.
b) The referee's remark that the QUAINT propagation scheme leads to weaker constraints is only correct, in fact, at high rigidity, while at low rigidity (where the different functional dependence allows QUAINT to best reproduce AMS02 antiprotons data, see e.g. Fig.12 of arxiv1906.07119) QUAINT leads to stronger constraints for low DM masses. This is clearly manifest in our fig. 4, right panel. We have now added a comment to clarify this aspect.

Answer to minor comments:

1- We agree with the referee that there are circumstances where Wilks theorem is not applicable, and indeed we had mentioned that. We had probably been too synthetic and, due to possible confusion on this point, we have expanded and clarified in Sec. 4 as well as Table 2. There are two reasons why Wilks theorem does not apply in our null hypothesis testing: The fact that the null hypothesis lies at the boundary of the parameter space, as mentioned by the referee, and the fact that the parameter m_chi is not defined under the null hypothesis. The first problem can be tackled with Chernoff’s theorem, indeed, and we now mention that. The second issue is related to correcting for trial factors, and requires numerical evaluation. Since we get test statistics values very similar to those of our Ref. [75], we chose to use their same convention (which is also used in 1903.02549 and in 1712.00002, to quote but a few) to recast likelihood ratio values into equivalent ’two-tailed' Gaussian significance. This also allows the reader to gauge how irrelevant global significances are (since they are computed in [75]). Now this is clarified.

We note however that the referee's remark does not apply to our 95\% CL setting procedure, which never involves a comparison with the null hypothesis. We are aware that alternative conventions for setting limits exist, such as Eq. (14) in arxiv:1007.1727, but this is not amounting to use Chernoff’s theorem, rather to adopt a different test-statistic than the one used here (our eq.s 5-8). Since our limit-setting (correspondence of the 95\% CL to a Delta chi^2 of 3.84) is consistent with the TS introduced in eq. 5-8, and since it is not rare in the literature either (see for instance arxiv:1610.03071), we stick to this practice.

2- The captions of figures 3 and 6 have been complemented to deal with the referee's remark. The author names in the legends of these figures are now associated to bibliographic references in the captions.

3- We are unsure of what has prompted this remark by the referee, but here are some clarifications:
a) If wondering about the rationale to choose BIG or SLIM (they would be similar) as benchmark, this is based on the performances of these two models in fits to CR data (notably secondaries over primaries) where BIG/SLIM tend to outperform QUAINT. b) We are not sure either of why the referee thinks that choosing a different benchmark would ease comparison with the literature. If we limit ourselves to results of recent papers reported e.g. in Fig. 3 and Fig. 6, all of them use a diffusion coefficient with a low rigidity behaviour following a broken power law in rigidity (as we do with BIG), with the exception of [Giesen et al. 15], by a subgroup of us, which pre-dates the modern analyses justifying the BIG/SLIM model; and [Cui et al. 16], which instead choose to break the injection power law. Both reasons thus seem to comfort us in our choice to refer to BIG, and we prefer to keep it.

4- As customary in the literature, in our model-independent approach we consider only kinematically open annihilation channels, i.e. only on-shell annihilation products. The tools that we use, in addition, do not include the off-shell option. We have added a comment in Section 2.1 (just below eq. (1)) to better specify this point.

---

## Round 2 · Referee Report · Anonymous (Referee 2) · 2022-4-22

Report

I would like to thank the authors for the various additions to the manuscript, which in my opinion constitute significant improvements, in particular for non-expert readers. The new version offers a pedagogical and thorough discussion of the best practices in the analysis of anti-proton data, which will be very influential and useful for the community. I am happy to recommend publication, but cannot resist leaving a few more comments: - I find the arguments of the authors against profiling over nuisance parameters quite convincing. However, my personal conclusion from this discussion is that a good compromise could be obtained by marginalizing (rather than profiling) over nuisance parameters. It would be interesting to understand whether this leads to similar results as the procedure currently implemented. - I agree with the authors that the LR hypothesis test that they perform is both common and reasonable. I just wanted to point out that there are reasons to suspect that a MC simulation of mock experiments would lead to somewhat different p-values. - A small comment that slipped through the original review: As far as I am aware, the Neyman-Pearson lemma only applies to simple hypotheses (with no free parameters), i.e. it only covers the case of a likelihood ratio, not a profile likelihood ratio. Finally, I would like to thank the authors for the additional explanation regarding the different propagation schemes, which I found very illuminating.

Requested changes

No changes to the manuscript required.

---

## Round 2 · Author Response

Please see the responses to the reports.

---

## Round 2 · List of Changes

Please see the attached a PDF version with modifications in color.

---

## Editorial Decision

published